# Non-Amontons frictional behaviors of grain boundaries at layered material interfaces

Yiming Song [1,4], Xiang Gao [2,4], Rémy Pawlak [1], Shuyu Huang [1,3], Antoine Hinaut [1], Thilo Glatzel [1], Oded Hod [2] ✉, Michael Urbakh [2] ✉ & Ernst Meyer [1] ✉

Against conventional wisdom, corrugated grain boundaries in polycrystalline graphene, grown on Pt(111) surfaces, are shown to exhibit negative friction coefficients and non-monotonic velocity dependence. Using combined experimental, simulation, and modeling efforts, the underlying energy dissipation mechanism is found to be dominated by dynamic buckling of grain boundary dislocation protrusions. The revealed mechanism is expected to appear in a wide range of polycrystalline two-dimensional material interfaces, thus supporting the design of large-scale dry superlubric contacts.

Structural superlubricity (SSL), the fascinating phenomenon of ultra-low interfacial friction, originating from effective lateral force cancellation at crystalline interfaces, has evolved over the past two decades from being a purely theoretical concept to the verge of becoming of practical use[1]. Among various candidates for SSL realization, two-dimensional (2D) material interfaces demonstrate extraordinary potential, benefitting from unique weak van der Waals interlayer interactions accompanied by strong covalent intralayer networks inherent to 2D materials[2]. Since the 2004 milestone experimental verification of SSL in twisted nanoscale graphitic contacts[3], extensive scientific exploration of the scaling-up of SSL has been triggered. Notably, recent experimental studies have pushed the limit up to the micrometer-scale, based on single-crystal 2D material interfaces[4–6]. However, volume preparation of large-scale high-quality single-crystal 2D samples remains a challenging task under standard laboratory conditions[7,8]. At increasing length-scales, 2D materials, often synthesized via chemical vapor deposition (CVD) or pyrolysis, typically exhibit a polycrystalline structure consisting of misoriented crystal surface patches separated by grain boundaries (GBs)[9–15]. The former enhances interfacial incommensurability favoring SSL[16], whereas the latter may introduce additional energy dissipation channels that enhance friction, therefore challenging the scaling-up of SSL towards the macroscale.

GBs are typically formed through a chain of dislocations (e.g., pentagon-heptagon pairs in graphene) at the borderline of contacting misoriented grains. For 2D materials, the introduction of GBs leads to substantial modifications in their structural[11,13], mechanical[17–19], chemical[20], thermal transport[21–23], electrical[24–31], and ferromagnetic properties[32]. Recent theoretical and computational studies on the tribological properties of GBs in 2D layered interfaces have predicted unique frictional mechanisms involving a shear-induced GB protrusion (un)buckling mechanism that may lead to negative friction coefficients (NFCs)[33,34]. Furthermore, intricate moiré superstructure stick-slip dynamics and scattering over elongated GBs was predicted to enhance friction at high normal loads[35]. Going beyond single GB considerations, enhanced interfacial friction at multi-grain contacts was also predicted[36–38]. In accordance, recent experimental evidence indicate the enhancement of both van der Waals[39] and Coulombic[40] friction over 2D material GBs. This calls for an experimental investigation of the microscopic mechanisms underlying GB friction aiming to identify routes to control, manipulate, and eliminate it.

In this work, we investigate the mechanisms of 2D GB friction via detailed atomic force microscopy (AFM) experiments, rationalized by fully atomistic simulations and phenomenological modeling. Considering the prototypical polycrystalline graphene (PolyGr) system, we demonstrate that corrugated GBs present NFCs and non-monotonic velocity dependence of friction. Conversely, flat GBs are shown to exhibit linear friction increase with normal load, obeying Amontons' law, as well as logarithmic velocity dependence, similar to single-crystalline surfaces. Our atomistic simulations indicate that dynamic

[1]Department of Physics, University of Basel, Basel, Switzerland. [2]Department of Physical Chemistry, School of Chemistry, The Raymond and Beverly Sackler Faculty of Exact Sciences and The Sackler Center for Computational Molecular and Materials Science, Tel Aviv University, Tel Aviv, Israel. [3]Key Laboratory for Design and Manufacture of Micro-Nano Biomedical Instruments, School of Mechanical Engineering, Southeast University, Nanjing, China. [4]These authors contributed equally: Yiming Song, Xiang Gao. ✉e-mail: odedhod@tauex.tau.ac.il; urbakh@tauex.tau.ac.il; ernst.meyer@unibas.ch

snap-through GB protrusion (un)buckling mechanism plays a key role, allowing the construction of a phenomenological two-state model that fully rationalizes the experimental results.

## Results and discussion

To perform the experiments, single layered PolyGr films were grown on Pt(111) surfaces by CVD under ultrahigh vacuum (UHV) conditions. Non-contact (NC) and contact AFM measurements were used to perform in-situ investigations under UHV conditions (base pressure $\leq 1 \times 10^{-10}$ mbar) of the dissipative frictional behavior of elongated GBs formed between adjacent PolyGr grains of different lattice orientations (for a schematic representation see Fig. 1a and for further experimental details see "Methods" and Supplementary Notes 1-2).

The *in-situ* bimodal NC-AFM measurements provide the surface topography and the atomically resolved structure of the grown PolyGr. A typical corrugated GB of misfit angle (i.e., the relative lattice misorientation between two neighboring grains) of $\theta_{GB} = 21.43 \pm 0.66°$ is shown in Fig. 1b, c. The misfit angle is determined by fast Fourier transform (FFT) of the atomically resolved structures in Fig. 1c (see Supplementary Note 3 for further details of GB angle determination), which are superimposed on the moiré superstructures appearing due to the lattice and orientational mismatch between the graphene grains and the underlying Pt(111) substrate. The chain of GB dislocations clearly appears in the topography (Fig. 1b) and torsional oscillation frequency shift (Fig. 1c) maps as upward protrusions of average height of ~0.7 Å, in agreement with previous theoretical predictions[33] and experimental observations[13]. Notably, this GB defect corrugation is ~3.5 times larger than that of the observed moiré superstructures and its lateral extent of ~1 nm covers several carbon rings, making it clearly distinct from its surroundings.

A rough estimation of the spatially resolved energy dissipation map appears in Fig. 1d. This map is obtained from the beam deflection traces recorded during the non-contact topography scan of Fig. 1b (See "Methods" for further details). Already at this resolution, pronounced energy dissipation features are clearly observed above the GB protrusions. Higher resolution results, obtained from the torsional

frequency shift map of Fig. 1c, are presented in Fig. 1e. These results demonstrate that energy dissipation is localized around specific GB protrusions, suggesting that these defects are actively undergoing buckling/unbuckling transitions[33,34]. The double peak structure characterizing the energy dissipation trace (see inset in Fig. 1e) signifies three sliding steps across the GB: (i) first peak—downward buckling of the protrusion as the tip approaches the GB; (ii) region between the peaks—tip sliding over the suppressed protrusion with lower energy dissipation; and (iii) second peak—unbuckling of the protrusion as the tip leaves the GB region. Away from the GB, over the bulk area of the adjacent grains ultralow energy dissipation is observed in line with recent experimental observations[41,42].

To investigate the effect of external load on the frictional properties of GBs, we turned to perform frictional measurements under UHV conditions in the contact AFM mode. Figure 2a, b present lateral force maps of an extended GB having a misfit angle of $\theta_{GB} = 2.35 \pm 0.10°$. The latter is extracted using an FFT analysis of the atomic lattice orientations in the bordering grains (see Supplementary Fig. 4) that manifest different moiré superstructures. The clear stick-slip features with atomic periodicity measured across the grain regions indicate the pristine nature and degree of cleanness of the grown graphene samples. At the contact mode, the convolution between the AFM tip geometry (radius of curvature < 7 nm) and the surface topography results in artificially widened GB features that are imaged with an apparent width of ~6.83 ± 0.36 nm. Nonetheless, the lateral force map presents well-defined periodic patterns along the main axis of the GB with a period of $D = 5.34 \pm 0.23$ nm (see Fig. 2a, b), which are associated with individual GB dislocations. According to Frank's equation[10,43]:

$$\theta_{GB} = 2 \arcsin \frac{\left| \vec{b}_{(1,0)} \right|}{2D}, \tag{1}$$

where $\vec{b}_{(1,0)}$ is the Burgers vector of the most common edge-sharing heptagon-pentagon pair dislocation ($\left| \vec{b}_{(1,0)} \right| = 2.46$ Å, see Supplementary Fig. 5a), this periodicity corresponds to a misfit angle of

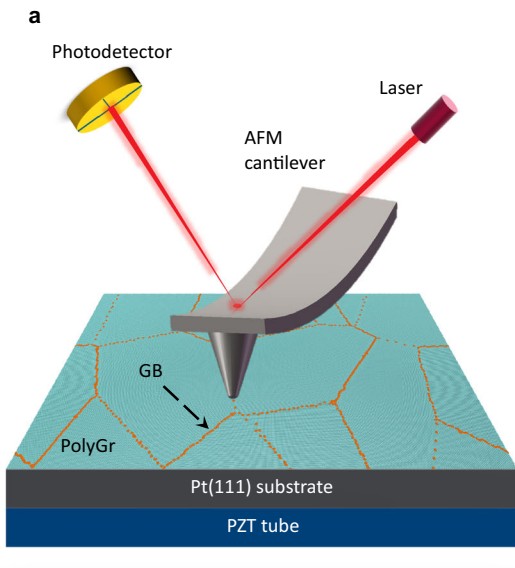

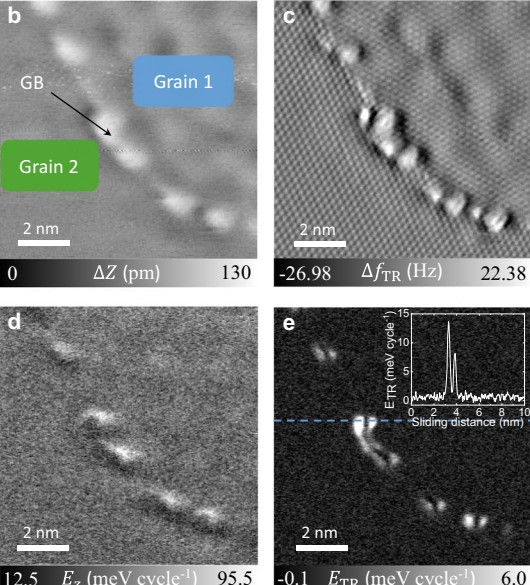

**Fig. 1 | Characterization and non-contact friction measurements of grain boundaries. a** Schematic diagram of the experimental set-up to characterize PolyGr grain boundaries (GBs). **b** Topography and (**c**) torsional frequency shift ($\Delta f_{TR}$) maps of a GB of misfit angle $\theta_{GB} = 21.43 \pm 0.66°$ over a scan area of 10×10 nm² obtained at the second flexural frequency shift of −340 Hz. **d-e** Vertical and torsional energy dissipation maps corresponding to panels b and c, respectively. The inset of panel e shows a cross-section of the energy dissipation map along the scan-line marked by the dashed blue line. In these measurements, the amplitudes of the second flexural and the torsional modes were $A_{2^{nd}} = 600$ pm and $A_{TR} = 80$ pm, respectively.

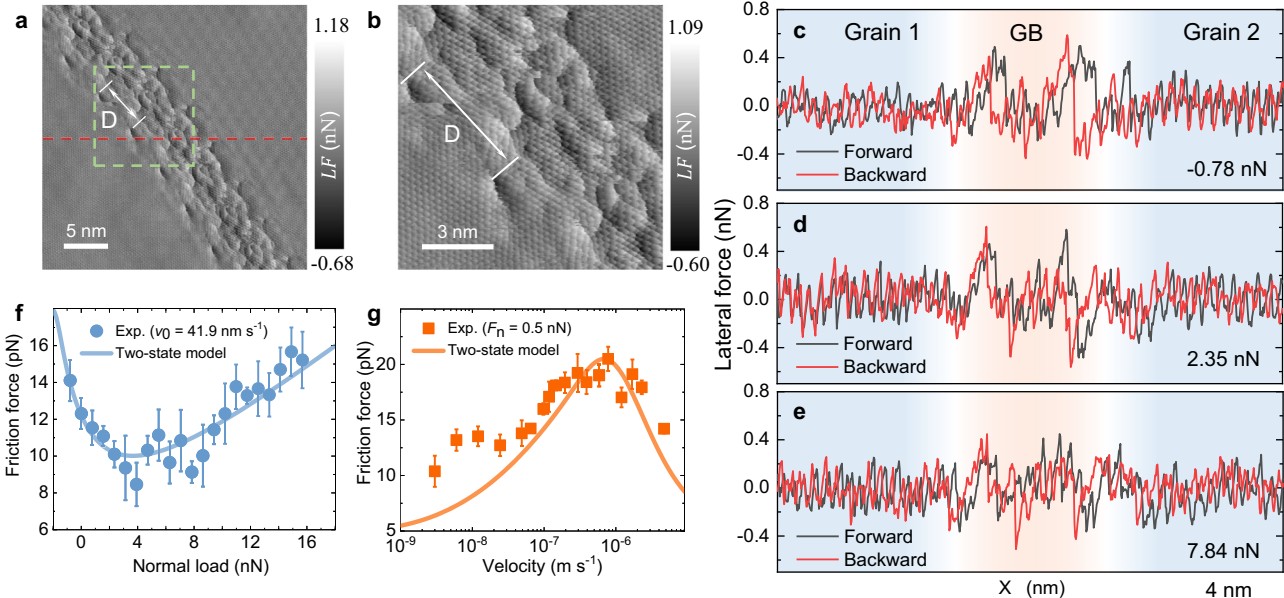

**Fig. 2 | Contact friction measurements of a corrugated GB. a** Lateral force map for a GB with a misfit angle of $\theta_{GB} = 2.35 \pm 0.10°$, measured under a normal load of 3.1 nN with a sliding velocity of 73.2 nm · s$^{-1}$. $D$ denotes the distance between neighboring GB dislocations. **b** Zoom-in on the area marked by the green dashed square in panel (**a**) showing atomically resolved lateral force patterns. **c-e** Lateral force loops taken across the red dashed line appearing in panel (**a**) under normal loads of (**c**) −0.78 nN, (**d**) 2.35 nN, and (**e**) 7.84 nN, where forward and backward traces are marked in black and red, respectively, and the GB region is marked by the light-red background. **f** Load dependence of the friction force (blue circles) averaged over 3–5 independent scans of an area of $30 \times 30$ nm$^2$ at a sliding velocity of 41.9 nm · s$^{-1}$. The error bars represent the corresponding standard deviations. **g** Velocity dependence of the average friction force (orange rectangles) measured under a normal load of 0.5 nN. The full lines in panels f and g represent the results of the two-state phenomenological model with the following parameters: $T = 300$ K, $E_1 = 0.18$ eV, $E_2 = 0.26$ eV, $\Delta x = 10.8$ Å, $\alpha = 0.2$ eV · GPa$^{-1}$, $\beta = 0.2$, $c_0 = 0.05$ eV, $N = 1$, $f_0 = 16.76$ kHz, $c_1 = 4.5$ pN, $\mu = 6 \times 10^{-4}$. Here, the effective protrusion stiffness is calculated as $k_0 = \frac{E_1 + E_2}{\Delta x^2}$, reflecting the fact that the maximum elastic energy stored by the spring $\left(\frac{1}{2}k_0\Delta x^2\right)$, cannot exceed $\Delta E_{max}$.

$\theta_{GB} \approx 2.64°$. The good agreement of this value with the independent FFT estimation based on the atomic lattice orientations, validates our GB characterization (see further validation in Supplementary Note 4 and Supplementary Fig. 5b) and confirms that each periodic GB pattern in the force map designates an isolated (1,0) type pentagon-heptagon dislocation, as $D$ matches the theoretical periodicity of (1,0) dislocations along the GB[10].

Figure 2c–e presents three force trace loops taken along the red dashed line appearing in Fig. 2a at increasing normal loads. Regardless of the value of the normal load, the average differences between the trace and retrace curves at the two grain regions are 3.56 and 6.68 pN, respectively, indicating very small energy dissipation due to non-conservative frictional forces. A significantly larger average difference of 42.54 pN is obtained over the GB region at a negative normal load of −0.78 nN (transferred via the inherent adhesion), signifying enhanced frictional energy dissipation. Notably, when increasing the external normal load to 2.35 and 7.84 nN, the average differences between the trace and retrace curves at the GB region reduce. Figure 2f presents the friction force as a function of normal load evaluated under a sliding velocity of 41.9 nm · s$^{-1}$ by averaging the lateral force traces over a square area of $30 \times 30$ nm$^2$ around the GB under a given normal load. While this area includes not only the GB itself but also part of the moiré superstructures of the adjacent grains, the latter, as shown above, have minor contribution to the overall results (see Supplementary Fig. 11)[41,42]. Notably, the average friction force (averaged over 3 to 5 independent scans) measured over the GB is found to decrease by up to a factor of 2 with increasing applied load, then it levels off at a normal load of ~4 nN, corresponding to a pressure of 1–2 GPa. The system therefore manifests a negative friction coefficient of −1.11 × 10$^{-3}$. Above a normal load of 8 nN, the friction force turns

to exhibit typical Amontons-like behavior with linear increase of friction with the normal load up to the highest load considered. A similar behavior is found for other corrugated GBs (see Supplementary Note 6).

As shown in Fig. 2g, at the regime of NFCs, we also found an atypical non-monotonic behavior of the friction with the sliding velocity, where, for example, the friction force peaks at a velocity of ~600 nm · s$^{-1}$ under a normal load of 0.5 nN. Overall, a 2-fold enhancement of the friction is found across three decades of the velocity increase. Conversely, in the plateau load regime of Fig. 2f, the friction velocity dependence exhibits a monotonic (logarithmic) increase with sliding velocity (see Supplementary Fig. 15).

These behaviors are qualitatively different from that previously found over moiré superstructures at surface grain regions, where low and nearly constant friction (< 10 pN) was observed at low normal loads and sliding velocities, followed by a linear or logarithmic increase, respectively, above a system-dependent threshold[41,42]. This suggests that different mechanisms underlay frictional energy dissipation at GBs and grain moiré regions.

A question arises whether the unique frictional behavior exhibited by corrugated GBs characterizes also flat GBs formed by a continuous chain of dislocations[10,11,33]. To address this question, we repeated our measurements for the flat GB with a misfit angle of $\theta_{GB} = 27.30 \pm 1.07°$, shown in the low temperature scanning tunneling microscopy (STM) images of Supplementary Figs. 9, 10. The lateral force map in Fig. 3a demonstrates three distinct regions that exhibit substantially different force traces, including two grains of different moiré superstructures and a continuous GB. Notably, the friction force measured over the GB is comparable to that measured over the grain of larger moiré supercell, rather than its small moiré tile counterpart. This suggests that flat GBs do not introduce additional energy dissipation channels not observed over moiré grain regions.

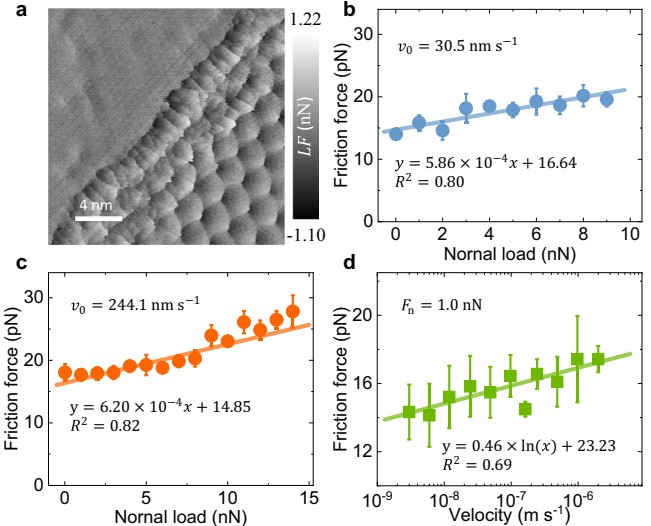

**Fig. 3 | Contact frictional behavior of a flat graphene GB. a** Lateral force map of a flat graphene GB with a misfit angle of $\theta_{GB} = 27.30 \pm 1.07°$, measured under a normal load of 5.7 nN and a sliding velocity of 146.5 nm · s⁻¹. **b, c** Load dependence of the frictional force measured at sliding velocities of 30.5 and 244.1 nm · s⁻¹, respectively. **d** Velocity dependence of the friction force measured under a normal load of 1.0 nN. The green squares represent the friction force. The error bars in panels (**b–d**) designate the corresponding standard deviations obtained by performing 5 to 6 independent scans. The solid lines are linear fits against the experimental data.

Figure 3b,c present the load dependence of the averaged GB friction measured at sliding velocities of 30.5 and 244.1 nm · s⁻¹, respectively, showing a linear kinetic friction force increase with normal load up to 14 nN, and kinetic friction coefficients lower than $1 \times 10^{-3}$, well within the superlubric regime. The slight deviation from linearity apparent in Fig. 3c may be attributed to the emergence of moiré-level friction over the large moiré supercell grain[41,42]. For this flat GB system, the commonly observed monotonic (logarithmic) increase of friction with sliding velocity is also obtained (see Fig. 3d). Recent experiments on flat MoS₂ GBs presented similar Amontons' law type friction force dependence on the applied normal load further supporting our findings[40]. We note that the values of the friction force measured for the flat GB (Fig. 3) are somewhat higher than those measured for the corrugated counterpart shown in Fig. 2. This comparison, however, is misleading since these two independent measurements have been performed with different AFM tips. Hence, the analysis focuses on the load dependence of friction and not on the absolute friction force values.

To rationalize the experimental findings regarding the unconventional load dependence of friction at corrugated GBs, we performed fully atomistic simulations of a model system consisting of a diamond tip sliding over a corrugated graphene GB (GB protrusion height of ~2 Å, see Fig. 4a) supported by a Pt(111) substrate. For comparison, we also performed simulations on a flat GB setup (GB protrusion height of < 0.2 Å, see Fig. 4b). The left grain of the PolyGr layer was oriented roughly in alignment with the $\langle 1\bar{1}0 \rangle$ lattice direction of the Pt(111) surface, leading to a large moiré period of ~2.2 nm. To form the corrugated or flat GBs, the right grain was rotated counterclockwise by misfit angles of $\theta_{GB} = 2°$ and 27.8°, respectively, yielding

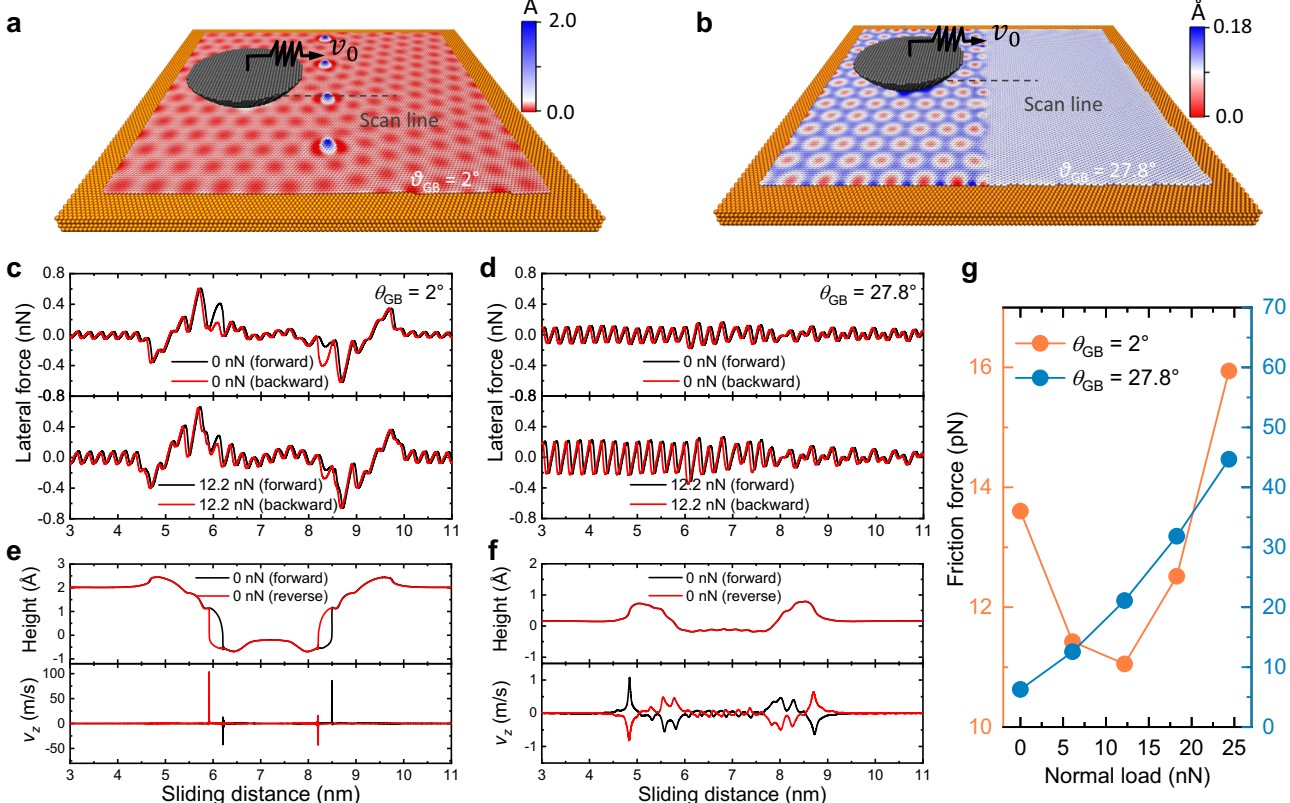

**Fig. 4 | MD simulations. a, b** Simulation setup for sliding over (**a**) a corrugated GB and (**b**) a flat GB. The gray and orange spheres represent the diamond tip and the Pt(111) substrate, respectively. The PolyGr atoms are colored according to their height above the average surface (see false color bars to the right of each panel). The lateral dimensions of the Pt(111) substrate are 41.6 × 40.8 nm². **c, d** Lateral force trace loops obtained under normal loads of 0 and 12.2 nN for (**c**) the corrugated and (**d**) flat GBs. **e, f** GB atom height and vertical velocity ($v_z$) trajectories for the (**e**) corrugated and (**f**) flat GBs. **g** The averaged GB friction as a function of normal load for the corrugated (orange) and flat (blue) GBs.

moiré periods of ~2 nm in the former case and on the order of the atomic lattice period or below for the latter. These model systems aim to mimic the experimental topographies presented in Figs. 2a and 3a. The sliding simulations were performed at zero temperature and at a sliding velocity of 2 m · s⁻¹ (See "Methods" and Supplementary Note 7 for further details).

The sliding simulation results reveal that the GB topography plays a vital role in the frictional behavior. Under zero external normal load, in addition to the atomic scale stick-slip motion, the corrugated GB exhibits significant differences between the forward and backward lateral force traces when the tip crosses a GB protrusion (see Fig. 4a, c). Increasing the normal load to 12 nN significantly diminishes these differences, in qualitative agreement with the experimental results presented in Fig. 2c–e. For the flat GB, only atomic-scale stick-slip motion is observed with no significant trace differences in the GB region (see Fig. 4d). Figure 4e presents the height and vertical velocity variations of a given atom, residing at a corrugated GB protrusion area, as a function of tip displacement. The sharp features characterizing the two traces clearly indicate a shear-induced buckling/unbuckling energy dissipation mechanism, similar to that recently predicted for multi-layered PolyGr interfaces[33,34]. As the tip approaches an upward protrusion, it gradually presses down on it until a sudden snap-through event occurs, resulting in a downward protruding state. When the tip leaves the GB area, the protrusion buckles back to its upward protruding state (See Supplementary Movie 1). The instantaneous buckling velocity reaches up to ~100 m · s⁻¹, giving rise to considerable kinetic energy dissipation in the vertical direction. As the normal load increases, the snap-through buckling behavior is suppressed, the features in the height and velocity profiles become smoother, and the overall energy dissipation decreases (See Supplementary Movie 2 and Supplementary Fig. 18).

In contrast, for the flat GB, where effective cancellation of lateral strain between contacting dislocations results in negligible out-of-plane corrugation, the transition (un)buckling energy barrier is small[33]. This results in much smoother height and vertical velocity trajectories with out-of-plane velocity variations below 1 m · s⁻¹ and significantly reduced energy dissipation (see Fig. 4f and Supplementary Movie 3).

The relations between the averaged friction force and the normal load for the corrugated and the flat GBs are presented in Fig. 4g, showing non-monotonic dependence with NFCs in the low load regime for the corrugated GB, and linear growth with load for the flat GBs, both of which are in good qualitative agreement with the experimental results shown in Figs. 2f, and 3b, c (see Supplementary Note 7 for details regarding the calculation of the average friction force). Our MD simulations reveal a similar qualitative non-Amontons behavior for other scanline directions and substrate thicknesses (see Supplementary Figs. 19, 20).

The fact that our non-reactive atomistic simulations correctly describe both the force traces and the experimentally observed dependence of friction on the normal load indicates that they capture well the underlying frictional mechanisms, thus providing a microscopic understanding of the involved phenomena. Specifically, this indicates that tip-surface bonding, expected to occur at higher normal loads for pristine surfaces[44], has minor effect on friction also in our case despite the expected higher reactivity of GB regions.

Nonetheless, due to the computational burden involved, such simulations are limited to sliding velocities considerably higher than those accessible in experiments. To address this issue, we harnessed the understanding gained by the atomistic simulations to devise a simplistic phenomenological model that captures the essential physical ingredients to describe the frictional behavior of corrugated GBs at a wide range of experimental conditions. The key ingredient of the phenomenological model is a shear-induced transition between two

(meta-)stable states, representing the upward and downward protruding GB dislocation configurations. A similar model was developed in details in Ref. 34 for a PolyGr contact embedded within a graphitic stack. In what follows, we provide a brief overview of a modified model focusing on its adaptation to the case of an AFM tip sliding over a corrugated layered material GB.

We mark by $\Delta E(x(t), \sigma)$ the transition energy barrier (TEB) between the two states. We assume that it depends mainly on the distance ($x(t)$) between the tip apex and the GB protrusion and on the normal load ($\sigma$). For simplicity, we assume linear spatial interpolation between the bare buckling energy barrier before the tip reaches the protrusion region, $\Delta E_{\max} = \Delta E(x = 0)$, and the barrier obtained when the tip resides above the center of the protrusion, $\Delta E_{\min}(\sigma) = \Delta E(x = \Delta x, \sigma)$. We note that $\Delta E_{\max}$ is independent of the normal load since the tip does not reside above the protrusion. We further assume the following linear relations between $\Delta E_{\max}$, $\Delta E_{\min}$ and $\sigma$:

$$\begin{cases} \Delta E_{\min}(\sigma) = \Delta E_{\min}(0) - \alpha\sigma \\ \Delta E_{\min}(0) = \beta\Delta E_{\max} - c_0 \end{cases}, \quad (2)$$

where $\alpha$, $\beta$, and $c_0$ are fitting parameters. In accordance with the simulation results, we also assume that the protrusions buckle and unbuckle independently, such that the survival probability of the system at a given state can be described by a first-order transition rate equation of the form:

$$\frac{dp(x(t), \sigma, \Delta E_{\max})}{dx} = \frac{dp(x(t), \sigma, \Delta E_{\max})/dt}{dx(t)/dt} = -\frac{f_0}{v_0} e^{\frac{-\Delta E(x,\sigma)}{k_B T}} p(x, \sigma, \Delta E_{\max}) \quad (3)$$

where $v_0 = dx(t)/dt$ is the constant sliding velocity of the tip, $f_0$ is the attempt frequency, which formally depends on the structure of the potential energy surface near equilibrium, and the exponential Arrhenius factor introduces the dependence of the transition rate on the barrier height and the thermal energy $k_B T$. Correspondingly, the probability density distribution of the protrusion to buckle at a given tip position is given by $f(x, \sigma, \Delta E_{\max}) = -dp(x, \sigma, \Delta E_{\max})/dx$. Given this probability density, we can now evaluate the elastic energy dissipated by a tip sliding induced buckling event of an individual protrusion over the sliding path $\Delta x$ via:

$$\Delta w(\sigma, \Delta E_{\max}) = \int_0^{\Delta x} dx \frac{k_0}{2} x^2 f(x, \sigma, \Delta E_{\max}) H(\Delta E(x, \sigma)) \quad (4)$$

Here, $k_0$ is an effective stiffness characterizing the elastic deformation at the tip-protrusion contact, such that the integrated term signifies the dissipated elastic energy invested in depressing the dislocation up to the buckling point when the tip is located at point $x$. The Heaviside step function screens unphysical negative TEBs.

To account for the fact that different GB protrusions can have different bare energy barriers, one should average the individual protrusion energy dissipation over the distribution of barrier heights, $P_b(\Delta E_{\max})$, yielding the following expression for the friction force:

$$F_{GB}(\sigma) \approx N \int \frac{\Delta w(\sigma, \Delta E_{\max})}{\Delta x} P_b(\Delta E_{\max}) d\Delta E_{\max}, \quad (5)$$

where $N$ is the total number of GB protrusions crossed simultaneously by the tip along a given scanline. For simplicity, we further assume a uniform distribution of the bare energy barriers bound to the energy range $[E_1 : E_2]$. Finally, to account for baseline friction resulting from, e.g., atomic-scale stick-slip dynamics, an additional friction term, $F_0$, is

added:

$$F_{tot}(\sigma) = F_{GB}(\sigma) + F_0, \tag{6}$$

where, per Amonton's friction law, $F_0$ is assumed to depend linearly on the normal force $F_n$:

$$F_0 = \mu F_N + c_1 \tag{7}$$

While a logarithmic dependence of $F_0$ on velocity may be also expected, previous experimental results demonstrated that it remains constant in the low velocity superlubric regimes considered in our experiments[41,42]. For a direct comparison with experimental results, we further assume that the contact area has a circular shape with a constant radius of 12 Å (i.e., 4.5 nm² in area, close to that of the tip in the MD simulations) to convert normal pressure, $\sigma$, to the normal force, $F_n$. This is in line with previous theoretical predictions[45] and supported by our MD simulations (Supplementary Note 7).

The model parameters are extracted from our MD simulation results (whenever possible) or fitted against the experimental measurements, within reasonable physical bounds[33,34,46] (see caption of Fig. 2 and Supplementary Note 8). The resulting parametrized two-state model reproduces well the unique load and velocity dependence of the friction force across corrugated GBs demonstrated in our experiments (see Fig. 2f, g) and atomistic simulations (see Fig. 4g and Supplementary Note 8). The phenomenological model allows us to identify the dominating factors responsible for the observed frictional behavior of corrugated GBs. Specifically, the NFC behavior can be attributed to the lowering of the buckling energy barrier with increasing tip normal load, leading to a decrease in the dissipated energy per buckling event.

The non-monotonic velocity dependence of the friction force is traced to balancing two competing effects: (i) decrease of thermally assisted buckling probability with increasing velocity due to the reduced time that the tip spends over the GB protrusion; and (ii) increase in the overall dissipated energy per buckling event, resulting from the fact at higher sliding velocities the tip can shift further along the sliding path before overcoming the energy barrier, such that buckling may occur at larger protrusion depressions.

The observations of unconventional frictional properties of negative differential friction coefficients and non-monotonic velocity dependence are not limited to the corrugated GB considered above and are well reproduced in other corrugated GBs (see Supplementary Figs. 12, 13). This demonstrates the general nature of our findings, which have a significant impact on scaling-up structural superlubricity towards macroscopic contacts that inevitably involve polycrystalline layered material interfaces. Assuming a constant GB density, one might naïvely conclude that overall GB friction contribution would grow linearly with contact area, thus eliminating structural superlubricity at large scales. Our results, however, demonstrate that by harnessing the unconventional frictional properties of GBs (e.g., negative friction coefficients and non-monotonic velocity dependence) together with other unique control schemes, such as gate-tunable behavior[40], one may restore and control structural superlubricity in large-scale polycrystalline two-dimensional material interfaces.

## Methods
### Sample preparation
The PolyGr layer was grown in a UHV chamber on a freshly prepared Pt(111) surface. Prior to the chemical vapor deposition, the Pt(111) surface was cleaned by cycles of sputtering and high-temperature annealing. PolyGr was synthesized by means of ethylene dosing directly onto the hot surface (see Supplementary Note 1 for a detailed description).

### UHV AFM measurements
The homebuilt ultrahigh vacuum atomic force microscope of beam deflection type (base pressure of $<1\times10^{-10}$ mbar) was operated at room temperature (300 K) with a Nanonis Control System by SPECS GmbH. The bimodal mode was used to characterize the PolyGr surface and detect non-contact friction. Contact mode was used to conduct friction force measurements, where the AFM tip is in direct contact with the PolyGr layer.

### Energy dissipation in non-contact friction measurements
The energy dissipation (in units of eV per oscillation cycle) was calculated according to the following formula[47,48]:

$$E_{T-S} = E_0 \left( \frac{V_{exc}}{V_{exc,0}} - \frac{f}{f_0} \right), \tag{8}$$

where $E_0$ is the energy loss per oscillation cycle caused by the intrinsic dissipation of the freely oscillating cantilever, $V_{exc}$ is the voltage needed to maintain a constant excitation amplitude, $f$ is the eigenfrequency of the cantilever, and $V_{exc,0}$ and $f_0$ are the corresponding values for the free cantilever. Here, the fix-ended cantilever is driven by a shaking piezo-element, the exciting voltage of which was modulated to maintain constant oscillation amplitude when interacting with the PolyGr sample.

In the beam deflection NC-AFM measurements, the second flexural oscillation mode (resonant frequency of $f_{2nd} = 1.02$ MHz and amplitude of $A_{2nd} = 600$ pm) was used to control the tip-sample distance. The relatively large oscillation amplitude allows us to obtain a consistent value of the average tip-sample distance and thus a clear topographic map, against which the corresponding dissipation map can be compared in the NC mode. This implies that the tip is strongly influenced by long-range dissipative interactions other than those induced by the GB itself. While such interactions reduce the contrast of the NC dissipation map, the high sensitivity of $V_{exc}$ and $f$ towards the tip-sample distance allows to obtain a meaningful image.

To analyze the NC-AFM friction signal at the torsional mode, the cantilever was oscillated at the torsional resonant frequency with an excitation amplitude of $A_T = 80$ pm $\ll A_{2nd}$. In such case, the energy loss signal is more sensitive to the local interactions between the tip and the PolyGr surface, thus reducing other long-range interaction effects. Therefore, the energy dissipation map provided by tortional mode measurements (Fig. 1e) shows a much higher contrast compared to the beam deflection map (Fig. 1d).

### Atomistic simulations
The molecular dynamics (MD) simulation model system consisted of a spherical-cap shaped diamond tip (2.3 nm in height, 5 nm in radius at the cut surface) sliding atop a bi-crystalline graphene layer supported by a 1.36 nm thick Pt(111) substrate, as shown in Fig. 4a, b. The PolyGr atomic arrangements were created using a Voronoi tessellation approach developed by Shekhawat et al[12,49]. The inter-atomic interactions within the diamond tip and the graphene layer were described with the second-generation reactive empirical bond order (REBO) potential[50]. The inter-atomic interactions within the Pt substrate were described via the embedded-atom-method (EAM) potential[51]. Due to the lack of dedicated anisotropic force-fields for the PolyGr/Pt(111) and PolyGr/diamond interfaces, we adopted the isotropic Lennard-Jones (LJ) potential to describe these interactions. While being unable to simultaneously capture both binding and sliding energy landscapes, with appropriate parameterization this potential was shown to provide a good qualitative description of frictional processes[41,42,52]. The LJ parameters for the diamond/graphene carbon atom interactions were taken to be $\sigma_{CC} = 3.4$ Å and $\varepsilon_{CC} = 0.00284$ eV[53]. For carbon/Pt interactions, we benchmarked the LJ parameters against available DFT and experimental reference values for the binding energy and equilibrium

interlayer distance between graphene and Pt(111), yielding $\sigma_{C-Pt} = 3.35$ Å and $\varepsilon_{C-Pt} = 0.006$ eV, which were then used for both the tip-substrate and the graphene-substrate interactions (See Supplementary Note 7).

All sliding simulations were performed at zero temperature by driving the diamond tip via a spring with stiffness of 10 N · m$^{-1}$ at a constant velocity of 2 m · s$^{-1}$ under external normal loads in the range of 0-25 nN. All simulations were carried out using the LAMMPS package[54]. See Supplementary Note 7 for further simulation details.

## Data availability
The data that support the findings presented in this paper are available within the paper and its supplementary information. A source data file is provided with this paper and can be downloaded at: https://doi.org/10.5281/zenodo.13768451.

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

## Acknowledgements

Y.S. acknowledges fruitful discussions with M. Kisiel. The Basel group thank the Swiss Nanoscience Institute (SNI) and the European Research Council (ERC) under the European Union's Horizon 2020 research and innovation program (ULTRADISS grant agreement No 834402). E.M., T.G., and R.P. acknowledge the Swiss National Science Foundation (grant no. 200021_228403 and 200021L_219983). X.G. acknowledges the postdoctoral fellowships of the Sackler Center for Computational Molecular and Materials Science and Ratner Center for Single Molecule Science at Tel Aviv University. M.U. acknowledges the financial support of the Israel Science Foundation, grant No. 1141/18 and the ISF-NSFC joint grant 3191/19. O.H. is grateful for the generous financial support of the Israel Science Foundation under grant No. 1586/17, the Heinemann Chair in Physical Chemistry, Tel Aviv University Center for Nanoscience and Nanotechnology, and the Naomi Foundation for generous financial support via the 2017 Kadar Award.

## Author contributions

E.M., M.U., and O.H. conceived the original idea behind this study. Y.S. and E.M. designed the experimental aspects of the study, and Y.S. performed the experiments and analyzed the experimental data. S.H. and T.G. assisted with the friction measurements. R.P. and A.H. assisted with the LT-STM characterizations. M.U., O.H., and X.G. designed the simulations and analyzed their results. X.G. conducted the simulations and theoretical calculations. E.M., M.U., O.H., X.G., and Y.S. wrote the manuscript. All the authors commented on the manuscript.

## Competing interests

The Authors declare no competing interests.
