## [Transparent Peer Review file · Nature Communications]

Non-Amontons Frictional Behaviors of Grain Boundaries at Layered Material Interfaces

Corresponding Author: Professor Oded Hod

Version 0:

Reviewer comments:

Reviewer #1

(Remarks to the Author)

This is a joint experiment-theory paper about the friction of a nanotip sliding on a deposited graphene monolayer with different types of grain boundaries (GBs). The point of the work is to demonstrate that severely buckled GBs unexpectedly exhibit a decreasing friction under weak increasing load— a behaviour dubbed non-Amontons. In that regime there is also a non-standard velocity weakening of friction above a critical velocity. The observation is rationalised through modelling and simulations connecting these anomalies with the tip-induced unbuckling/re-buckling of the GB. In agreement with that, the non-Amontons effects disappear for unbuckled GBs.

The work is interesting and in my view worthy of publication. However, the paper contains many obscure points, which raise questions that need to be addressed and clarified.

Main questions.

1. The non-Amontons effect is demonstrated only for exceedingly broad, buckled and apparently structured GBs. First, a clearer explanation ought to be given about the nature and origin of these broad apparent structures. Second, it is not clear if the non-Amontons effect is caused by mere buckling, be it narrow or broad, or just by its large width as in Fig. 2 a-b. To clarify that, the authors should show the behaviour of friction on a narrow buckled GB, such as that of Fig. 1 b-e. Another information required is the friction dependence upon the sliding direction, since the effective GB width crossed by the tip will change relative to the tip diameter depending on the sliding direction.

2. The non-Amontons part of sliding friction is attributed, as illustrated by the simulations, to a load-induced reduction of the GB buckling magnitude. The corresponding unbuckling operated by the front edge of the tip, followed by re-buckling behind the trailing edge becomes therefore less important at larger loads. If the nonmonotonic friction decrease at large velocities was accordingly attributed to the inability of the GB to unbuckle and re-buckle fast enough, then the typical crossing time below which this would occur should be $t=2R/v$, where R is the tip radius and v the sliding velocity. With the present experimental values, one would estimate $t \sim 10^{-3}$ s., an extremely long time for a nanoscale event at room temperature. That is hard to understand and should be explained— together with the choice of the many parameters provided in Fig.2's caption.

3. The buckling/unbuckling friction contribution over massively buckled GBs is proposed to be additional to the standard, regular Amontons part. Yet, the friction of Fig. 3, where the GB is unbuckled, is larger than that of Fig.2, where the GB is buckled. That inconsistency should be explained. In addition, the authors should also quantitatively describe the friction over a perfect, GB free area.

4. The MD results should be connected with the theory. Conversely, how close are quantities obtained by fitting, to real experimental ones?

Minor questions.

5. Frank's Eq. (1) is presented. What role does parameter D play in experiment and/or theory?

6. In Eq. (6) one assumes a constant friction F_0 to account for the baseline friction. The dependence of F_0 on load and velocity should be clarified.

7. There is almost no difference between supplementary movies 1 and 2. Adding some explanations or color bars might be helpful.

Reviewer #2

(Remarks to the Author)

The manuscript of Song et al. reports a combined experimental/simulation study of the friction between a tip and polycrystalline graphene supported by Pt(111). For small misfit angles between two graphene grains strong buckling of the grain boundary (GB) and a nonmonotonic friction versus normal force relationship with negative differential friction coefficients for small normal loads are observed in the Basel UHV friction force microscope experiment, while for large misfit angles buckling is reduced and differential friction coefficients are positive over the whole load range. These results are rationalized within an atomistic model of a diamond tip sliding over a small and large misfit angle graphene GB on Pt(111). A qualitative agreement between experiment and the atomistic simulations is observed although the sliding velocities differ by many orders of magnitude. The negative differential friction is explained by a buckling instability: less elastic energy is stored in the GB protrusion for higher loads i.e. less energy is lost for higher normal forces. Similar simulations for a graphene stack with GBs have been published by some of the authors in a previous Nature Communications article. In this article, a simplified model is described that captures the buckling instability by stress assisted transitions in a two state model. The beauty of this previous work is that accompanying atomistic calculations were reproduced by extracting many of the model parameters from the atomistic model. In the current manuscript, the two state model parameters are treated as fitting quantities. Good agreement between experiment and simulations can be achieved by tuning the parameters. The authors claim that the fit parameters converge to physically reasonable values.

Although the mechanism has already been published in Nat. Comm. as a theoretical prediction, the current manuscript has some elements of novelty:

- 1) In the modified atomistic simulation model, the buckling mechanism occurs also on Pt(111).
- 2) Combined with the simulations the experiments suggest that the buckling plays a role in real 2D materials.

It is not my task to decide whether this is novel enough for a Nature subjournal, but I'm convinced that this study should be published. However, there is some room for improvement for the simulation part (see below) and I believe that the manuscript would benefit from a major revision.

A) The atomistic model geometry consists of a 1.3 nm thick Pd slab covered by the graphene. The maximum applied load results in pressures in the GPa range. Obviously, the elastic response of the Pd substrate is quite unrealistic. Thus, it should be checked that the friction curves in Fig. 4g show still the same behavior if the thickness of the Pd slab is increased to a size such that the Hertzian pressure maximum is within the sample (usually the slab should be thicker than 3-5 times the tip radius)

B) The tip is modelled as a diamond spherical cap while in the experiments an Si tip is used. What is the structure of the surface after etching away the SiO₂? Is there any passivation or can we assume a reactive Si tip in the experiment? Why is the tip in the simulation non-reactive?

C) For reactive tips, other processes could come into play in the experiments such as intermittent formation of covalent bonds between the corrugation and the Si tip - see e.g. Phys. Rev. Res. 5, L012049 (2023). The strong corrugation at the GB leads to partially sp³ hybridized states indicating to some degree de-passivation of the graphene. This possibility should be discussed and the authors should find arguments why this can be excluded in the models.

D) The simplified two state model contains many fit parameters and one cannot deduce its validity from the fact that these parameters can be tuned to reproduce the experimental friction-load curves. The authors claim that the parameters stay within reasonable physical boundaries. I think many readers would feel more comfortable when the authors could extract as many of the parameters as possible from the atomistic model. For instance E_1 , E_2 , Δx and k_0 could be measured in the atomistic model [graphene on Pd(111)]. A comparison with the atomistic Fig. 4g could be used to check the accuracy of the parameters. Such a multiscale model that uses information from full atomistic models to parametrize rate models would greatly improve the manuscript. Maybe the agreement with the experiment would be less perfect, but the comparison would be more stringent and meaningful.

E) Why is a constant contact area assumed in the two state model? In the atomistic model a spherical cap tip is used and assuming a Hertz contact model would result in a contact area that increases with load.

Version 1:

Reviewer comments:

Reviewer #1

(Remarks to the Author)

The authors made a serious effort to clarify their points in their response and in the revised manuscript. I find the outcome satisfactory, and suggest publication in present form.

Reviewer #2

(Remarks to the Author)

I am satisfied with the authors' response to my critique and recommend publication of the revised manuscript.

RESPONSE TO REVIEWERS' COMMENTS

We would like to thank the reviewers for their thorough evaluation of our manuscript and for their constructive comments that helped us to improve our paper. In the following response, we fully address all remarks and questions raised by the reviewers and provide a detailed account of the corresponding modifications made to the text. For clarity, we reproduce the reviewers' comments in italic followed by our detailed response in boldface text.

Reviewer #1 (Remarks to the Author):

“This is a joint experiment-theory paper about the friction of a nanotip sliding on a deposited graphene monolayer with different types of grain boundaries (GBs). The point of the work is to demonstrate that severely buckled GBs unexpectedly exhibit a decreasing friction under weak increasing load— a behaviour dubbed non-Amontons. In that regime there is also a non-standard velocity weakening of friction above a critical velocity. The observation is rationalised through modelling and simulations connecting these anomalies with the tip-induced unbuckling/re-buckling of the GB. In agreement with that, the non-Amontons effects disappear for unbuckled GBs.

The work is interesting and in my view worthy of publication. However, the paper contains many obscure points, which raise questions that need to be addressed and clarified.”

Response

We thank the referee for his/her positive evaluation of our manuscript.

“Main questions.

1. The non-Amontons effect is demonstrated only for exceedingly broad, buckled and apparently structured GBs. First, a clearer explanation ought to be given about the nature and origin of these broad apparent structures. Second, it is not clear if the non-Amontons effect is caused by mere buckling, be it narrow or broad, or just by its large width as in Fig. 2a-b. To clarify that, the authors should show the behaviour of friction on a narrow buckled GB, such as that of Fig. 1b-e. Another information required is the friction dependence upon the sliding direction, since the effective GB width crossed by the tip will change relative to the tip diameter depending on the sliding direction.”

Response:

We thank the referee for raising this important point. In fact, the typical structure of a graphene GB consists of a chain of decoupled or coupled 5-7 rings of width of ~ 1 nm regardless of GB angle (See Fig. 4a-b and S5a). The apparently different widths in Fig. 1b-e and Fig. 2a-b are caused by the different experimental characterization methods. In particular, Fig. 1b-e represents the atomic resolution characterization of GB by bimodal non-

contact (NC) atomic force microscopy (AFM) measurements. In this technique, the frequency shift of the torsional oscillation Δf_{TR} is measured, while controlling the tip-sample distance via the frequency shift of the second flexural oscillation. Since only the apex tip atoms interact strongly with the sample and experience attractive forces during scanning (see *Noncontact Atomic Force Microscopy: 3rd edition*, 2015, P. 4), high-resolution imaging can be achieved, yielding an accurate width estimation for the measured GBs. On the contrary, the lateral force measurements in Fig. 2a-b were conducted using the contact AFM mode. When the tip is in mechanical contact with the sample, the measured lateral force map reflects a convolution between the AFM probe geometry and the GB topography (see *Fundamentals of Friction and Wear on the Nanoscale*, 2015, P. 162) which apparently widens the measured GBs features. Therefore, despite the apparent differences, the GBs presented in Fig. 1c-e and 2a-b are actually of similar widths, and present similar non-Amontons behavior.

Regarding the effect of scanline, Figs. 2 and S12-S13 of the original text, present friction force measurements performed along different scan line directions with respect to the GBs, showing a similar qualitative non-Amontons behavior.

To further evaluate the dependence of friction force on the sliding direction, we performed additional MD simulations along an inclined scanline of 15 degree with respect to the normal of the GB. The force traces presented in Fig. R1a show significant energy dissipation at the GB, associated with strong buckling of GB protrusions and non-Amontons frictional behavior (friction reduction with increased normal load), as shown in Fig. R1b, both of which are consistent with the results obtained for the original scanline.

To clarify this point, we have made the following modifications in the revised text:

On page 11 of the revised main text, the sentence “Our MD simulations reveal a similar qualitative non-Amontons behavior for other scanline directions and substrate thicknesses (see Figs. S19-S20).” has been added.

Furthermore, Fig. R1 has been added as Fig. S19 in the revised SI along with a relevant discussion.

Fig. R1. MD simulation results for a $\theta_{GB} = 2^\circ$ GB obtained along a 15° scanline direction with respect to the normal of the GB. (a) Lateral force trace loops obtained under normal loads of 0 and 12.2 nN; (b) GB atom height and velocity trajectories under zero normal load.

2. The non-Amontons part of sliding friction is attributed, as illustrated by the simulations, to a load-induced reduction of the GB buckling magnitude. The corresponding unbuckling operated by the front edge of the tip, followed by re-buckling behind the trailing edge becomes therefore less important at larger loads. If the nonmonotonic friction decrease at large velocities was accordingly attributed to the inability of the GB to unbuckle and re-buckle fast enough, then the typical crossing time below which this would occur should be $t=2R/v$, where R is the tip radius and v the sliding velocity. With the present experimental values, one would estimate $t \sim 10^{-3}$ s., an extremely long time for a nanoscale event at room temperature. That is hard to understand and should be explained— together with the choice of the many parameters provided in Fig.2's caption.

Response:

We thank the reviewer for raising this important point. The millisecond time-scale that the referee refers to corresponds to the time separation between the buckling event due to tip compression and the unbuckling event due to tip departure of the GB region. However, the separate buckling and unbuckling nanoscale events themselves occur at a much shorter time scale of 10^{-11} sec (*Nat. Commun.* 12, 5694 (2021)), as shown by our atomistic simulations. This time scale is considerably shorter than the typical response time of the AFM apparatus' estimated to be ~ 16.8 kHz (the corresponding value obtained for the former fitting was ~ 21 kHz) by the fitted attempt frequency f_0 (see SI section 8.1), which is in good agreement with previous friction force microscopy experiments (see, e.g., *Phys. Rev. Lett.* 91, 084502 (2003); *Phys. Rev. Lett.* 99, 166102 (2007)). Hence, one cannot measure individual buckling events using standard AFM approaches.

Regarding the choice of model parameters in Fig. 2g: the bounds of bare buckling transition energy barriers ($E_1 = 0.18$ eV and $E_2 = 0.26$ eV), as well as the rate of reduction of the transition barrier with normal load ($\alpha = 0.2$ eV/GPa) are estimated from the instant kinetic energy pulses produced following GB buckling during the sliding simulations, as shown in Fig. R2. The characteristic sliding distance ($\Delta x = 10.8$ Å) matches the typical GB width (~ 1 nm). Assuming that the elastic energy stored in the effective spring (mimicking the GB in our phenomenological model) cannot exceed the transition energy barrier, we estimate the effective spring stiffness, k_0 , via the relation $\frac{E_1+E_2}{2} = \frac{1}{2}k_0\Delta x^2$. The value of the number of GB protrusions influenced by the tip, ($N=1$ or 2) is calculated from the ratio between the tip radius and the inter-protrusion separation, D . As suggested by Reviewer #1 and to obtain a better fit to the experimental results over the high normal load regime (see Fig. 2 of the revised main text), instead of being constant F_0 is now assumed to be linearly dependent on the normal force F_n , $F_0 = \mu F_n + c_1$, as suggested by Amonton's friction law (see Fig. R5). The remaining parameters ($\beta = 0.2$, $c_0 = 0.05$ eV, $\mu = 6 \times 10^{-4}$ and $c_1 = 4.5$ pN) are fitted to obtain good agreement between the model and simulation results. Notably, the fitted friction coefficient, μ , lies as expected in the superlubric sliding regime.

Fig. R2. (a) Kinetic energy profiles calculated for a $\theta_{GB} = 2^\circ$ GB system under various normal loads. (b) Kinetic energy pulse intensity as a function of normal load.

To clarify this point, we have made the following modifications in the revised text:

On page 13 of the revised main text, the sentence “By fitting the model parameters (within reasonable physical bounds^{33,34,45} - see caption of Fig. 2) against the experimental results” has been revised to “The model parameters are extracted from our MD simulation results (whenever possible) or fitted against the experimental measurements, within reasonable physical bounds^{33,34,45} (see caption of Fig. 2 and Supplementary Sec. 8.1). The resulting parametrized two-state model reproduces well the unique load and velocity dependence of the friction force across corrugated GBs demonstrated in our experiments (see Figs. 2f, g) and atomistic simulations (see Fig. 4g and Supplementary Sec. 8.2).”. Accordingly, the sentence “We note that under an external load of 8 nN, the phenomenological curve deviates from the experimental results (see Fig. 2f), due to the emergence of a different

frictional mechanism (i.e., enhanced Pauli repulsions), not accounted for by the former, but captured by our atomistic simulations (see Fig. 4g).” was omitted.

Correspondingly, Sec. 8.1 with an explanation in line with the above has been added to the revised SI.

3. The buckling/unbuckling friction contribution over massively buckled GBs is proposed to be additional to the standard, regular Amontons part. Yet, the friction of Fig. 3, where the GB is unbuckled, is larger than that of Fig.2, where the GB is buckled. That inconsistency should be explained. In addition, the authors should also quantitatively describe the friction over a perfect, GB free area.

Response:

We thank the referee for raising this point. In the present work, the typical friction force of corrugated GB obtained in Fig. 2g is 18.4 ± 0.9 pN under a normal load of 0.5 nN and a sliding velocity of 195.3 nm/s, which is comparable to the value of 18.1 ± 1.4 pN measured for the flat GB shown in Fig. 3 under similar conditions (normal load of 0 nN and sliding velocity of 244.1 nm/s). Nonetheless, the quantitative comparison between the absolute friction forces in these two cases is irrelevant due to the use of different tips in the two independent measurements. Notably, what is important is the qualitatively different response of the measured friction towards external load in the two cases.

For friction over a pristine graphene, the typical atomic scale friction is <10 pN and remains mostly constant under low normal load and sliding velocity.

To clarify this point, we have made the following modifications in the revised main text:

On page 7 of the revised main text, the sentence “These behaviors are qualitatively different from that previously found over moiré superstructures at surface grain regions, where low and nearly constant friction was observed at low sliding velocities, followed by a logarithmic increase above a system-dependent threshold velocity.⁴¹” has been revised to “These behaviors are qualitatively different from that previously found over moiré superstructures at surface grain regions, where low and nearly constant friction (<10 pN) was observed at low normal loads and sliding velocities, followed by a linear or logarithmic increase, respectively, above a system-dependent threshold.^{41,42}”

On page 9 of the revised main text, the sentence “We note that the values of the friction force measured for the flat GB (Fig. 3) are somewhat higher than those measured for the corrugated counterpart shown in Fig. 2. This comparison, however, is misleading since these two independent measurements have been performed with different AFM tips. Hence, the analysis focuses on the load dependence of friction and not on the absolute friction values.^{41,42}” has been added.

4. The MD results should be connected with the theory. Conversely, how close are quantities obtained by fitting, to real experimental ones?

Response:

As detailed in our response to point #2 raised by the referee, the model parameters were extracted from the MD simulations either directly from the structural properties of the GB or from the dynamical behavior of the system under load and shear stresses induced by the tip. This includes the bare potential energy barrier bounds E_1 , E_2 ; the rate of barrier reduction under load, α ; the characteristic GB width, Δx ; the effective protrusion vertical stiffness, k_0 ; and the attempt frequency, f_0 , which is consistent with previous experimental values (*Phys. Rev. Lett.* 91, 084502 (2003)). Finally, all other two-state model parameters (β , c_0 , μ , and c_1) are fitted against the experimental results, also providing good agreement with the atomistic MD simulations.

The corresponding modifications are detailed in our response to point #2 above.

Minor questions.

5. Frank's Eq. (1) is presented. What role does parameter D play in experiment and/or theory?

Response:

The parameter D denotes the separation distance between neighboring GB dislocations. It affects the mutual strain cancellation between neighboring dislocations, the protrusion height, and the buckling energy barrier. A larger inter-GB distance leads to higher buckling energy barrier bounds (E_1 , E_2) but a smaller number of GB protrusions (N) simultaneously affected by the sliding tip.

6. In Eq. (6) one assumes a constant friction F_0 to account for the baseline friction. The dependence of F_0 on load and velocity should be clarified.

Response:

We thank the reviewer for raising this point. Considering F_0 originates from the atomic lattice friction over pristine graphene surface, F_0 is expected to follow a linear dependence on load and a logarithmic dependence on velocity. However, since this system is in the superlubric regime, the dependence of F_0 on load and velocity is insignificant in the low load and low velocity ranges characteristic of our experiments (*Nano Lett.* 22, 9529–9536 (2022); *Nano Lett.* 23, 10, 4693–4697, (2023)). Nonetheless, to fit the experimental results also in the higher load regime, F_0 is now assumed to linearly depend on the normal load. The fitted

friction coefficient, $\mu = 6 \times 10^{-4}$, clearly demonstrates that our system lies well within the superlubric regime.

To clarify this point, we have made the following modification in the revised main text:

On page 13 of the revised main text, the sentence “where, per Amonton’s friction law, F_0 is assumed to depend linearly on the normal force F_n ”:

$$F_0 = \mu F_n + c_1, \quad (1)$$

While a logarithmic dependence of F_0 on velocity may be also expected, previous experimental results demonstrated that it remains constant in the low velocity superlubric regimes considered in our experiments.^{41,42} has been added.

7. There is almost no difference between supplementary movies 1 and 2. Adding some explanations or color bars might be helpful.

Response:

We thank the reviewer for pointing out this issue. Movies 1 and 2 intend to demonstrate the very different cases with strong GB buckling under low load and weak GB buckling under high load, respectively. To improve the contrast, we have added color bars in each movie and emphasized the occurrence of (un)buckling events in the revised caption for Movie 1.

Reviewer #2 (Remarks to the Author):

The manuscript of Song et al. reports a combined experimental/simulation study of the friction between a tip and polycrystalline graphene supported by Pt(111). For small misfit angles between two graphene grains strong buckling of the grain boundary (GB) and a nonmonotonic friction versus normal force relationship with negative differential friction coefficients for small normal loads are observed in the Basel UHV friction force microscope experiment, while for large misfit angles buckling is reduced and differential friction coefficients are positive over the whole load range. These results are rationalized within a atomistic model of a diamond tip sliding over a small and large misfit angle graphene GB on Pt(111). A qualitative agreement between experiment and the atomistic simulations is observed although the sliding velocities differ by many orders of magnitude. The negative differential friction is explained by a buckling instability: less elastic energy is stored in the GB protrusion for higher loads i.e. less energy is lost for higher normal forces. Similar simulations for a graphene stack with GBs have been published by some of the authors in a previous Nature Communications article. In this article, a simplified model is described that capture the buckling instability by stress assisted transitions in a two state model. The beauty of this previous work is that accompanying atomistic calculations were reproduced by extracting many of the model parameters from the atomistic model. In the current manuscript, the two state model parameters are treated as fitting quantities. Good agreement between

experiment and simulations can be achieved by tuning the parameters. The authors claim that the fit parameters converge to physically reasonable values.

Although the mechanism has already been published in Nat. Comm. as a theoretical prediction, the current manuscript has some elements of novelty:

1) In the modified atomistic simulation model, the buckling mechanism occurs also on Pt(111).

2) Combined with the simulations the experiments suggest that the buckling plays a role in real 2D materials.

It is not my task to decide whether this is novel enough for a Nature subjournal, but I'm convinced that this study should be published. However, there is some room for improvement for the simulation part (see below) and I believe that the manuscript would benefit from a major revision.

Response

We thank the referee for his/her positive evaluation of our manuscript and for providing us with an opportunity to further improve our manuscript.

A) The atomistic model geometry consists of a 1.3 nm thick Pd slab covered by the graphene. The maximum applied load results in pressures in the GPa range. Obviously, the elastic response of the Pd substrate is quite unrealistic. Thus, it should be checked that the friction curves in Fig. 4g show still the same behavior if the thickness of the Pd slab is increased to a size such that the Hertzian pressure maximum is within the sample (usually the slab should be thicker than 3-5 times the tip radius)

Response:

We thank the reviewer for raising this important point. To check the effect of Pt slab thickness, we have repeated some of our simulations with a slab thickness of up to 13 nm, which is ~3 times the tip radius. Fig R3 shows the load dependence of the average friction force obtained for various slab model thicknesses. Clearly, the results converge already for a slab thickness of 6.5 nm. For all substrate thicknesses considered a similar nonmonotonic load dependence is obtained. For the 1.3 nm thick slab model, used to obtain the results presented in the main text, the minimum lies at a somewhat higher load and its value is somewhat larger. Nonetheless, the qualitative non-Amontons behavior remains the same.

To clarify this point, we have made the following modifications:

On page 11 of the revised main text, the sentence "Our MD simulations reveal a similar qualitative non-Amontons behavior for other scanline directions and substrate thicknesses (see Figs. S19-S20)." has been added.

Fig. R3 and a corresponding discussion have been added to Sec. 7.5 of the revised SI.

Fig. R3 . Effect of Pt substrate model thickness on the load dependence of the average GB friction.

B) The tip is modelled as a diamond spherical cap while in the experiments an Si tip used. What is the structure of the surface after etching away the SiO₂? Is there any passivation or can we assume a reactive Si tip in the experiment? Why is the tip in the simulation non-reactive?

Response

We thank the referee for raising this important point. The tips (PPP-CONT, Nanosensors) used in this work are manufactured with single crystal silicon, pointing into the <100> crystal direction (*Rev. Mod. Phys.* 75, 949 (2003)). Following an Ar⁺ sputtering process, the oxidized layer covering the tip apex is removed. Prior to the friction measurements, several scans encompassing a large surface area were conducted to locate the desired GB regions. During these scans ultra-low friction between tip and surface was measured (in the pN regime) and a very stable tip motion was observed over thousands of scanning cycles. These observations indicate that the tip remains chemically inert and physically stable during the friction experiments. Should chemical reactivity govern the sliding, one would expect the friction to rise towards the nN regime (*Phys. Rev. B* 60, R11301 (1999)). Furthermore, the fact that our non-reactive atomistic simulations correctly describe both the force traces and the experimentally observed dependence of friction on the normal load indicates that tip-surface bonding, expected to occur at higher normal loads for pristine surfaces, has minor effect on friction also in our case despite the expected higher reactivity of GB regions.

To clarify this point, we have made the following modifications:

On pages 11-12 of the revised main text, the paragraph “The fact that our non-reactive atomistic simulations correctly describe both the force traces and the experimentally observed dependence of friction on the normal load indicates that they capture well the underlying frictional mechanisms, thus providing a microscopic understanding of the

involved phenomena. Specifically, this indicates that tip-surface bonding, expected to occur at higher normal loads for pristine surfaces, has minor effect on friction also in our case despite the expected higher reactivity of GB regions.” has been added.

On page 15 of the revised SI, the sentence “Furthermore, the use of non-reactive dynamics, which excludes tip-substrate covalent bonding, is justified by the fact that our simulation results correctly describe both the force traces and the experimentally observed dependence of friction on the normal load.” has been added.

C) For reactive tips, other processes could come into play in the experiments such as intermittent formation of covalent bonds between the corrugation and the Si tip - see e.g. Phys. Rev. Res. 5, L012049 (2023). The strong corrugation at the GB leads to partially sp^3 hybridized states indicating to some degree depassivation of the graphene. This possibility should be discussed and the authors should find arguments why this can be excluded in the models.

Response:

We thank the reviewer for raising this point and providing the relevant reference. Notably, Ref. *Phys. Rev. Res.* 5, L012049 (2023) indicates that bonding between a SiO_2 tip and a graphene surface occurs at contact pressures exceeding 10 GPa, which is several times higher than the normal load range considered in the present work. While GB may indeed be more reactive due to local curvature effects, tip induced flattening recovers the sp^2 intralayer graphene bonding nature thus reducing its reactivity. As mentioned above, this is clearly indicated by our experimental findings. The fact that our non-reactive simulations are able to fully rationalize the experimental results, further supports our understanding that tip-surface bonding, if any, has minor effect on friction in the present case.

This point is clarified by the modifications detailed in the response to the previous point.

D) The simplified two state model contains many fit parameters and one cannot deduce its validity from the fact that these parameters can be tuned to reproduce the experimental friction-load curves. The authors claim that the parameters stay within reasonable physical boundaries. I think many readers would feel more comfortable when the authors could extract as many of the parameters as possible from the atomistic model. For instance E_1 , E_2 , Δx and k_0 could be measured in the atomistic model [graphene on Pd(111)]. A comparison with the atomistic Fig. 4g could be used to check the accuracy of the parameters. Such a multiscale model that uses information from full atomistic models to parametrize rate models would greatly improve the manuscript. Maybe the agreement with the experiment would be less perfect, but the comparison would be more stringent and meaningful.

Response:

We thank the reviewer for pointing out this critical point and providing the useful suggestions. While already in the original text we followed the two-state model parameters choice strategy suggested by the referee, we have opted to reexamine and refit the parameters against the MD simulation results. Specifically, the bounds of bare buckling transition energy barriers ($E_1 = 0.18$ eV and $E_2 = 0.26$ eV), as well as the rate of reduction of the transition barrier with normal load ($\alpha = 0.2$ eV/GPa) are now estimated from the instant kinetic energy pulses produced following GB buckling during the sliding simulations, as shown in Fig. R4. The characteristic sliding distance ($\Delta x = 10.8$ Å) is chosen to match the typical GB width (~ 1 nm). Assuming that the elastic energy stored in the effective spring (mimicking the GB in our phenomenological model) cannot exceed the transition energy barrier, we estimate the effective spring stiffness, k_0 , via the relation $\frac{E_1+E_2}{2} = \frac{1}{2} k_0 \Delta x^2$. The value of the number of GB protrusions influenced by the tip, ($N=1$ or 2) is calculated from the ratio between the tip radius and the inter-protrusion separation, D , and the attempt frequency, $f_0 = 16.76$ kHz, which corresponds to the characteristic frequency of the cantilever, is chosen to be consistent with previous experimental values (*Phys. Rev. Lett.* 91, 084502 (2003)). As suggested by Reviewer #1 and to obtain a better fit to the experimental results over the high normal load regime (see Fig. 2 of the revised main text), instead of being constant F_0 is now assumed to be linearly dependent on the normal force F_n , $F_0 = \mu F_n + c_1$, as suggested by Amonton's friction law (see Fig. R5). The remaining parameters ($\beta = 0.2$, $c_0 = 0.05$ eV, $\mu = 6 \times 10^{-4}$ and $c_1 = 4.5$ pN) are fitted to obtain good agreement between the model and simulation results. Notably, the fitted friction coefficient, μ , lies as expected in the superlubric sliding regime.

Fig. R4. (a) Kinetic energy profiles calculated for a $\theta_{GB} = 2^\circ$ GB system under various normal loads. (b) Kinetic energy pulse intensity as a function of normal load.

To clarify this point, we have made the following modifications in the revised text:

On page 13 of the revised main text, the sentence “By fitting the model parameters (within reasonable physical bounds^{33,34,45} - see caption of Fig. 2) against the experimental results” has been revised to “The model parameters are extracted from our MD simulation results

(whenever possible) or fitted against the experimental measurements, within reasonable physical bounds^{33,34,45} (see caption of Fig. 2 and Supplementary Sec. 8.1). The resulting parametrized two-state model reproduces well the unique load and velocity dependence of the friction force across corrugated GBs demonstrated in our experiments (see Figs. 2f, g) and atomistic simulations (see Fig. 4g and Supplementary Sec. 8.2).” Accordingly, the sentence “We note that under an external load of 8 nN, the phenomenological curve deviates from the experimental results (see Fig. 2f), due to the emergence of a different frictional mechanism (i.e., enhanced Pauli repulsions), not accounted for by the former, but captured by our atomistic simulations (see Fig. 4g).” was omitted.

Correspondingly, Sec. 8.1 with an explanation in line with the above has been added to the revised SI. Experimental results for additional GBs have also been refitted as shown in Figs. S12-S13 of the revised SI.

A comparison between the refitted model and the simulation results, is presented in Fig. R5 (also appearing in Sec. 8.2 of the revised SI) demonstrating a similar qualitative behavior with a vertical shift that may be attributed to the use of a simplified tip model in the atomistic simulations.

Fig. R5. Comparison between the load dependence of the friction force obtained using MD simulations (green circles) and the two-state model (blue line) for the $\theta_{GB} = 2^\circ$ system. The GB (black line) and pristine surface (red line) contributions to the two-state model are also presented.

E) Why is a constant contact area assumed in the two state model? In the atomistic model a spherical cap tip is used and assuming a Hertz contact model would result in a contact area that increases with load.

Response:

We thank the reviewer for raising this valid point. Our experimental system involves a nanometric single crystal silicon AFM probe (PPP-CONT, Nanosensors), pointing in the $\langle 100 \rangle$ direction (*Rev. Mod. Phys.* 75, 949 (2003)) and operating under relatively low normal load. Prior to the friction measurements, the native oxide had been etched away by Ar^+ sputtering, probably resulting in a stepped atomic structure at the tip apex. Under these conditions, the contact area is estimated (see *Nature* 435, 929 (2005)) to remain constant under loads below 12.53 nN, which is higher than the upper bound of our experimentally applied load (<10 nN) for observing non-Amontons behavior.

This is further supported by our atomistic MD simulation results that demonstrate a constant lateral deformation profile of the graphene surface under varying tip loads up to 12 nN (see Fig. R6).

To clarify this point, we have made the following modifications in the revised text:

On page 13 of the revise main text, the sentence “This is in line with previous theoretical predictions⁴⁴ and supported by our MD simulations (Supplementary Sec. 7.6).” has been added.

On page 22 of the revised SI, Fig R6 and a short discussion have been added as Sec. 7.6.

Fig. R6. MD simulation results of the deformation profiles of a polycrystalline graphene grain under different tip normal loads obtained using a 1.3 nm thick Pt substrate.

RESPONSE TO REVIEWERS' COMMENTS

We would further like to thank the reviewers for their thorough evaluation of our manuscript and for their recommendation for publishing our manuscript.

Reviewer #1 (Remarks to the Author):

The authors made a serious effort to clarify their points in their response and in the revised manuscript. I find the outcome satisfactory, and suggest publication in present form.

Response

We thank the reviewer for recommending our manuscript for publication.

Reviewer #2 (Remarks to the Author):

I am satisfied with the authors' response to my critique and recommend publication of the revised manuscript.

Response

We thank the reviewer for recommending our manuscript for publication.